# Enhancing Sample Diversity in Deep Convolutional GANs Using a Bayesian Framework

## Abstract

Generative Adversarial Networks (GANs) are proficient at generating synthetic data; however, they face the ongoing issue of mode collapse. This problem arises when the generator focuses on producing a limited set of data patterns that trick the discriminator during the optimization process, requiring novel solutions. In our endeavor to address mode collapse in Deep Convolutional Generative Adversarial Networks (DCGAN) and foster greater sample diversity, we introduce a Bayesian framework applied to DCGAN, referred to as Bayesian DCGAN. This framework makes three key contributions: (i) the integration of a weight distribution within the network, achieved through the application of the Bayes by Backprop method; (ii) employing a mean-field variational inference approach to approximate the posterior distributions of weights; and (iii) putting forth a mathematical approach to quantify the diversity present in the samples generated by Bayesian DCGAN, contrasting it with the output of conventional DCGAN. Our experimental results showcase that Bayesian DCGAN generates more diverse samples compared to its conventional counterpart, thereby significantly reducing uncertainty in neural networks. This enhancement in diversity is pivotal for creating robust and adaptable models, particularly in scenarios where a broader spectrum of data representations is essential for effective learning and generalization.

## 1 Introduction

Generative Adversarial Networks (GANs) (Goodfellow et al., 2014) have gained significant attention for their ability to transform white noise into realistic candidate samples using deep neural networks. The inherent adversarial process of GANs revolves around the interactive relationship between a generator ($G$) and a discriminator ($D$). The fundamental principle guiding the adversarial learning in GANs is rooted in the generator's endeavor to trick the discriminator by generating samples that follow the real distribution of data, $p_{data}$. Concurrently, the discriminator strives to distinguish between the artificially created samples by $G$ and the real ones originating from the actual data distribution. This competition is through a min-max game between $G$ and $D$ (Goodfellow et al., 2014):

$$\min_G \max_D V(G, D) = \mathbb{E}_{x \sim p_{data}(x)}[\log D(x)] + \mathbb{E}_{z \sim p_z(z)}[\log(1 - D(G(z)))] \tag{1}$$

A prominent variant of GAN, the Deep Convolutional Generative Adversarial Network (DCGAN) (Radford et al., 2015) architecture, was introduced to augment the internal complexity of both the generator and discriminator networks, and it serves as a foundational pillar in GAN research. Like traditional GANs, DCGAN consists of a discriminator ($D$) and a generator ($G$) sharing the same objective function. Both networks undergo an adversarial process, updating their weights based on calculated losses. This design employs convolution layers for both the generator (Figure 1) and discriminator networks, as opposed to the multilayer perceptrons used in the traditional GAN, which helps stabilize the training process of GANs (Radford et al., 2015; Farajzadeh-Zanjani et al., 2022).

GANs and DCGANs have significantly impacted the fields of artificial intelligence and machine learning (Saxena & Cao, 2021; Jabbar et al., 2021; Motwani & Parmar, 2020). It has been demonstrated that DCGAN

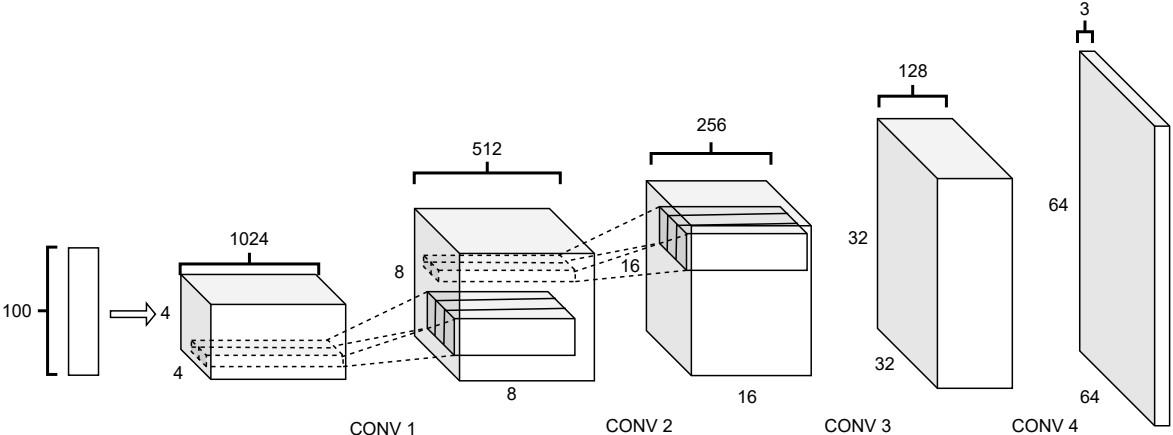

Figure 1: Generator architecture in DCGAN

is effective in addressing the challenges posed by unbalanced image datasets. This proves particularly valuable in the medical field, where detecting new emerging diseases often faces limitations due to insufficient datasets (Kazeminia et al., 2020; Bushra & Shobana, 2020; Frid-Adar et al., 2018a;b; Chuquicusma et al., 2018; Kitchen & Seah, 2017). GANs showed promising results in generating realistic images from poorly drawn forensic sketches for facial recognition applications (Bushra & Maheswari, 2021). It can be applied in the anomaly detection field (Schlegl et al., 2017), and has demonstrated remarkable results in representation learning (Lee et al., 2018; Mathieu et al., 2016). The development of GANs, especially with the advent of DCGANs, has opened new horizons for generating high-quality, realistic data, laying a foundation for future advancements in the field of generative modeling.

Despite the numerous benefits of GAN models (including DCGAN), they face several training challenges, such as vanishing gradients (Wiatrak et al., 2019), non-convergence (Lee & Seok, 2020), and mode collapse (Cao et al., 2018). If the discriminator becomes too good, the generator's training can stagnate due to vanishing gradients (Arjovsky & Bottou, 2017). Essentially, an overly optimal discriminator fails to provide the generator with sufficient feedback to improve its performance. On the other hand, as the generator improves during training, the discriminator's performance declines because it becomes harder to distinguish between real and fake data. If the generator becomes perfect, the discriminator's accuracy drops and makes random predictions. This creates a convergence issue for the GANs. DCGANs similar to GANs suffer from mode collapse (Durgadevi et al., 2021; Thanh-Tung & Tran, 2020). Mode collapse occurs when the generator concentrates on generating a restricted range of data patterns to mislead the discriminator. It fixates on a small number of predominant modes within the training data, falling short of capturing the complete diversity inherent in the data distribution. Consequently, the generator is prevented from exploring the entirety of the data distribution, resulting in the generation of repetitive data samples and overlooking the diverse variations present in the real dataset. Among these issues, mode collapse is particularly intriguing because it significantly limits the diversity of the generated samples. Addressing mode collapse is essential for improving the overall performance and robustness of GANs.

In this study, we address the problem of mode collapse in GANs by incorporating a Bayesian framework. Rather than using fixed weights in the network, we treat them as random variables with associated distributions. This approach allows the model to capture uncertainty and variability in its parameters, thereby facilitating the generation of more diverse samples. This framework makes three key contributions:

- Integrating a weight distribution within the network using the Bayes by Backprop method.

- Employing mean-field variational inference to approximate the posterior distributions of weights.

- Proposing a mathematical approach to quantify and compare the diversity of samples generated by Bayesian DCGAN with those from conventional DCGAN.

## 2 Related Work

To avoid mode collapse in GANs, various losses for discriminator have been proposed, such as f-GAN (Nowozin et al., 2016), least-squares (LSGAN) (Mao et al., 2017), Wasserstein distance (WGAN) (Arjovsky et al., 2017), and WGAN-GP (Gulrajani et al., 2017). Arjovsky et al. (2017) introduced the Wasserstein GAN (WGAN), a widely adopted metric that addresses this issue by enabling the discriminator to be trained to optimality without experiencing vanishing gradients. By enabling the discriminator to effectively learn and reject the generator's stable outputs, the generator is compelled to produce more diverse results. The WGAN proposes a new cost function based on the Wasserstein distance, also known as the Earth Mover's Distance, between probability distributions, which provides a smoother gradient throughout the training process. In WGAN, the discriminator (referred to as the critic) is required to be within the space of 1-Lipschitz functions, a condition enforced by the authors through weight clipping.

Saatci & Wilson (2017) introduces an innovative formulation for GANs employing Bayesian techniques. This method considers weight uncertainty in the construction of the GAN for data generation. To achieve this, the stochastic gradient Hamiltonian Monte Carlo (HMC) algorithm was employed as a sampling technique. It facilitates the marginalization of the posterior distributions over the weight parameters of both the generator $\theta_g$ and discriminator $\theta_d$. More specifically, it provides an approximation of the entire distribution over the parameters of both the generator and discriminator. Their findings illustrate that Bayesian GAN can effectively encompass a varied set of complementary representations of data.

Chien & Kuo (2019) proposes a Bayesian framework for GANs, drawing inspiration from GANs, Variational Autoencoders (VAE), and Bayesian neural networks. The introduced approach maximizes the variational lower bound of log likelihood within the GAN framework, resulting in a variational Bayesian GAN capable of exploring parameter posteriors and generating realistic synthesized samples. They applied the variational Bayesian inference for the posterior over parameter weights instead of using the stochastic gradient HMC algorithm for posterior sampling (see Figure 2). They argued that the variational Bayesian version of GAN demonstrated favorable performance in both unsupervised learning for a regression task and semi-supervised learning for a classification task, as evidenced by the experiments.

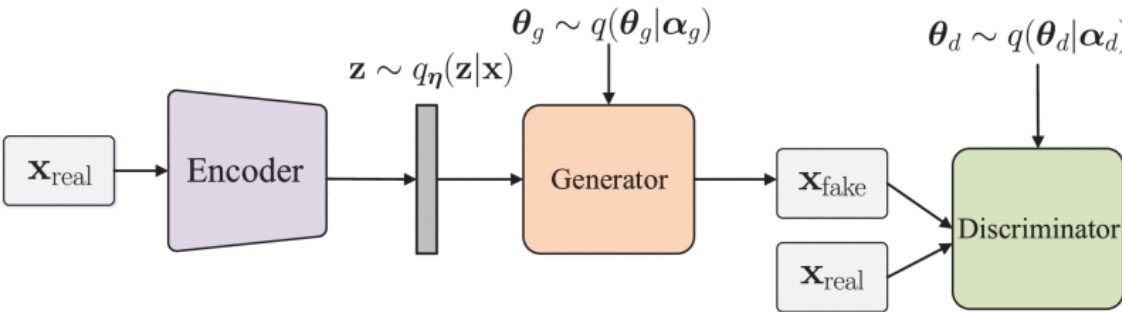

Figure 2: VGAN architecture presented by Chien & Kuo (2019)

In the present study, to address the issue of mode collapse in DCGANs, our approach introduces a Bayesian framework. This framework integrates a probability distribution over the weights within the network. We accomplish this by leveraging the Bayes by Backprop method, which employs mean field variational inference to approximate the posterior distributions of these weights (Blundell et al., 2015). In fact, instead of having fixed weights in the network, we treat them as random variables with distributions. This enables the model to capture uncertainty and variability in its parameters, which can help in generating more diverse samples. Moreover, we propose a mathematical method to quantify the diversity of the generated samples from our Bayesian DCGAN model compared to conventional DCGANs.

## 3 Method

In this section, we discuss the Bayesian Neural Network in 3.1, the application of mean-field variational inference in BNN in 3.2, and the Bayes by Backprop method as well as our proposed Bayesian DCGAN architecture in 3.3.

### 3.1 Bayesian Neural Network

Deep learning has gained significant attention across various fields but suffers from overfitting (Szegedy et al., 2013) and overconfident estimates (Goan & Fookes, 2020), necessitating careful management. A Bayesian Neural Network (BNN) deviates from the traditional deep neural network paradigm, offering a distinctive solution to the challenges posed by overfitting in deep learning models. In contrast to conventional neural networks that provide point estimates for weights, a BNN incorporates uncertainty into its predictions by representing the weights and biases as probability distributions instead of fixed values (Figure 3). This distinctive approach enables the incorporation of prior knowledge regarding these parameters into the model, allowing for the continuous refinement of beliefs as new data becomes available. By embracing probabilistic representations, Bayesian Neural Networks provide a flexible and adaptive framework, particularly useful for scenarios where uncertainty estimation is crucial. This design allows the model to not only make predictions but also express the range of potential outcomes, offering a more comprehensive understanding of the data and facilitating informed decision-making.

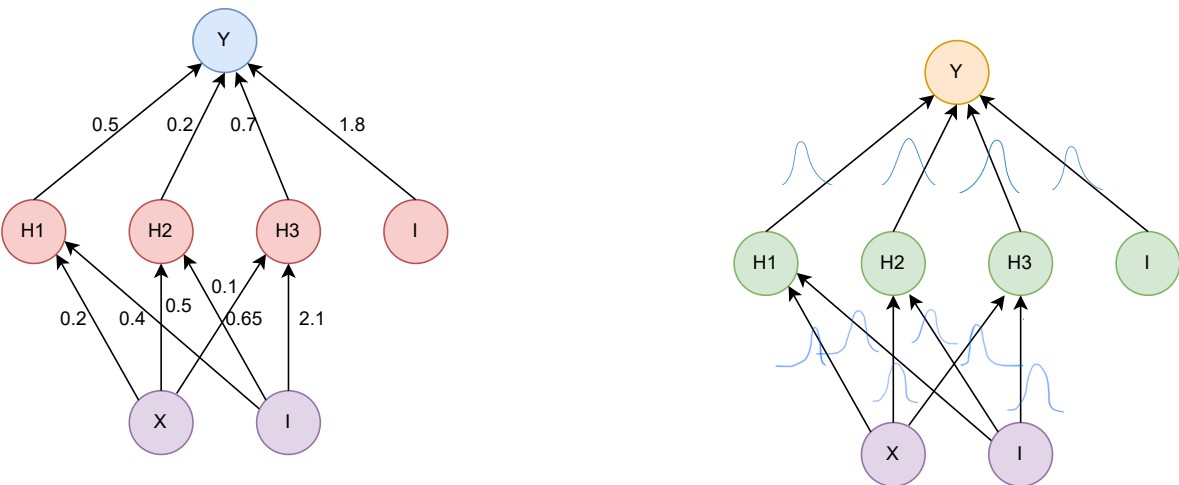

Figure 3: Bayesian Neural Networks (right) vs conventional Neural Networks

BNNs represent a unique class of stochastic neural networks that integrate Bayesian inference principles into their architecture. In the context of BNNs, network parameters, encompassing weights and biases,

are treated as random variables, and a probability distribution is systematically defined over them (Goan & Fookes, 2020). The distinguishing feature of BNNs lies in their capacity to learn and adapt to the distribution of weights. This learning process empowers BNNs to provide not just point estimates but a comprehensive measure of uncertainty regarding their predictions. This aspect of quantifiable uncertainty proves to be highly advantageous across a diverse array of applications.

Bayes' Theorem offers a powerful tool for representing a distribution over parameters conditioned on the observed data, denoted as the posterior probability distribution $P(W|D)$. The theorem is expressed through the equation:

$$P(W|D) = \frac{P(D|W)P(W)}{P(D)} = \frac{P(D|W)P(W)}{\int_{W'} P(D|W')P(W')\,dW'} \tag{2}$$

In this context, $P(W)$ represents the prior probability distribution, $P(D|W)$ denotes the data likelihood, and $P(D)$ is the marginal probability (also refers as normalization constant). This method of computing the posterior $P(W|D)$ is commonly referred to as exact inference. However, as indicated in the equation 2, computing $P(D)$ requires an integral over the weight space, which is often intractable. To overcome this challenge, two primary approaches have been proposed: Markov Chain Monte Carlo (MCMC) and Variational Inference. While Saatci & Wilson (2017) utilizes Hamiltonian Monte Carlo (an MCMC method), Blundell et al. (2015) employs the Variational Inference method to estimate the posterior distribution, a technique also employed in the current study and will be discussed in section 3.2.

### 3.2 Mean Field Variation Inference Method for BNN

Variational Inference is a method that seeks to approximate the posterior distribution $P(W|D)$ by introducing a surrogate distribution denoted as $q$. This surrogate, known as the variational distribution, is selected from a tractable family of distributions. The objective is to find the distribution $q$ that is the "closest" to the true posterior, thereby facilitating more manageable computations. In mean field variational inference, the algorithm operates under the assumption that the variational family is fully factorized. This means that the joint posterior distribution, denoted as $p(w|D)$, can be approximated by the product of individual variational distributions for each latent random variable. In other words, the approximation is expressed as:

$$p(w|D) \approx \prod_{i=1}^{n} q_i(w_i) \tag{3}$$

As previously stated, Variational Inference aims to identify the distribution $q(w)$ that best approximates the true posterior. The proximity between the two distributions is quantified using the Kullback-Leibler (KL) divergence (Kullback & Leibler, 1951; Joyce, 2011), denoted as $KL(q||p)$, defined as:

$$\text{KL}(q(w)||p(w|D)) = \int_w q(w) \log \frac{q(w)}{p(w|D)}\,dw = \mathbb{E}\left[\log \frac{q(w)}{p(w|D)}\right] \tag{4}$$

where $q(w)$ is the variational distribution and $p(w|D)$ is the true posterior distribution given observed data $x$. Also referred to as information gain, KL divergence measures the "information lost when $p(w|D)$ is approximated by $q(w)$". If the true posterior distribution $p(w|D)$ and the variational distribution $q(w)$ are identical, the KL divergence $KL(q(w)||p(w|D)))$ equals zero, indicating that no information is lost in the approximation. On the contrary, as $p$ and $q$ diverge, the value of $KL(q(w)||p(w|D))$ increases, signifying the growing difficulty in predicting the true distribution $p(w|D)$ based on the approximation $q(w)$. Minimizing the KL divergence corresponds to making the variational distribution $q(w)$ as close as possible to the true posterior $p(w|D)$. By expanding equation 4, we have:

$$
\begin{aligned}
\mathrm{KL}(q(w)||p(w|D)) &= \int_w q(w) \log \frac{q(w)}{p(w|D)} \, dw \\
&= \int_w q(w) \log \frac{q(w)p(w)}{p(w, D)} \, dw \\
&= \int_w q(w) \log \frac{q(w)}{p(w, D)} \, dw + \int_w q(w) \log p(D) \, dw \\
&= \int_w q(w) \log \frac{q(w)}{p(D|w)p(w)} \, dw + \log p(D) \\
&= \int_w q(w) \log \frac{q(w)}{p(w)} \, dw - \int_w q(w) \log p(D|w) \, dw + \log p(D)
\end{aligned}
\tag{5}
$$

In the above equation, $\int_w q(w) \log \frac{q(w)}{p(w)} \, dw - \int_w q(w) \log p(D|w) \, dw$ is negative of evidence lower bound (ELBO) (Kingma et al., 2019), so that:

$$
ELBO = \int_w q(w) \log p(D|w) \, dw - \int_w q(w) \log \frac{q(w)}{p(w)} \, dw
\tag{6}
$$

Considering equation 4, ELBO can be rewritten as follows:

$$
ELBO = \mathbb{E}_{w \sim q(w)} \left[ \log p(D|w) \right] - \mathrm{KL}(q(w)||p(w))
\tag{7}
$$

Therefore, minimizing $KL(q(w)||p(w|D))$ is equivalent to minimizing $-ELBO$, or maximizing $ELBO$.

### 3.3 Bayesian DCGAN Architecture

The Generative Adversarial Network utilized in the current study is specifically the DCGAN. DCGAN is a variant of GAN that exhibits a distinctive architecture where both the discriminator and generator incorporate convolutional layers. The use of convolutional layers is particularly advantageous as it allows the network to effectively capture spatial dependencies within the data. This spatial awareness significantly contributes to enhancing the overall quality of the generated images. In the endeavor to incorporate Bayesian Neural Networks (BNNs) into the framework of DCGAN, the methodology presented by Blundell et al. (2015) was followed. This approach involves treating the weights within the neural network as random variables and introducing a distribution over these weights. The motivation behind adopting this Bayesian perspective is to enable the model to account for uncertainty in its predictions, contributing to a more robust and versatile generative model.

A main aspect highlighted in Blundell et al. (2015) involves the utilization of the reparametrization technique. This technique plays a crucial role in ensuring that the variational parameters are sampled from a specific distribution, introducing a layer of stochasticity to the model. The assumption made in the study is that the variational posterior follows a diagonal Gaussian distribution. The process initiates by generating a sample from a unit Gaussian distribution, labeled as $\epsilon$. This sample undergoes a deterministic transformation, involving a shift by a mean $\mu$ and a scaling by a standard deviation $\sigma$, as specified in Equation 8. Consequently, this process yields the weight parameters $W$.

$$
W = \mu + \log(1 + exp(\rho)) \odot \epsilon
\tag{8}
$$

Here, $\mu$ represents the mean of the distribution, and $\rho$ is a parameter used to determine the standard deviation through the transformation $\sigma = \log(1 + \exp(\rho))$. Taking into account the shift and scaling operations, the weight parameters $W$ follow a normal distribution $\mathcal{N}(\mu, \sigma^2)$, where the set of learnable variational posterior parameters is represented as $\theta = (\mu, \rho)$. Thus, the variational posterior can be denoted as $q(w|\theta)$.

As previously mentioned, we adopt (similar to Blundell et al. (2015)) a Gaussian variational posterior. However, in terms of the prior, a scale mixture of two Gaussian densities with zero mean and distinct variances, where $\sigma_1 > \sigma_2$ and $\sigma_2 \ll 1$, is considered. The prior distribution is expressed as follows:

$$P(w) = \Pi_j \pi \mathcal{N}(w_j | 0,\ \sigma_1) + (1 - \pi)\mathcal{N}(w_j | 0,\ \sigma_2) \tag{9}$$

Here, $w_j$ represents the jth component of the weight vector $W$, and $\pi$ signifies the mixture weight, controlling the influence of each Gaussian component in the prior.

The objective function of Bayesian Neural Networks is to minimize KL divergence, according to Blundell et al. (2015) and the Equation 7, which can be rewritten as follows:

$$
\begin{aligned}
\text{KL}(q(w|\theta)||p(w|D)) &= -ELBO \\
&= \text{KL}(q(w|\theta)||p(w)) - \mathbb{E}_{w \sim q(w|\theta)}\left[\log p(D|w)\right] \\
&= \mathbb{E}_{w \sim q(w|\theta)}\left[\log \frac{q(w|\theta)}{p(w)}\right] - \mathbb{E}_{w \sim q(w|\theta)}\left[\log p(D|w)\right] \\
&= \mathbb{E}_{w \sim q(w|\theta)}\left[\log q(w|\theta)\right] - \mathbb{E}_{w \sim q(w|\theta)}\left[\log p(w)\right] - \mathbb{E}_{w \sim q(w|\theta)}\left[\log p(D|w)\right]
\end{aligned}
\tag{10}
$$

According to the Law of Large Numbers (LLN) (Uhlig, 1996), a fundamental statistical principle, $\lim_{n \to \infty} \frac{1}{n}\sum_{i=1}^{n} x_i = E(x)$ with a probability of 1. This law ensures that as the size of a sample increases, the sample mean converges towards the expected mean of the entire population. Utilizing LLN and drawing a sufficient number of Monte Carlo samples from the variational posterior $q(w|\theta)$, we can rewrite Equation 10 as follows:

$$\text{KL}(q(w|\theta)||p(w|D)) = \frac{1}{n}\sum_{i=1}^{n}\left(\log q(w^{(i)}|\theta) - \log p(w^{(i)}) - \log p(D|w^{(i)})\right) \tag{11}$$

where $w^{(i)}$ represents the ith Monte Carlo sample drawn from the variational posterior $q(w|\theta)$.

In the present study, we build upon the foundational concepts of Bayesian neural networks introduced by Blundell et al. (2015) by integrating them into a Convolutional 2-dimensional (Conv2D) and Convolutional 2-dimensional transpose (Conv2D Transpose) architecture to develop our Bayesian DCGAN model. This innovative approach allows us to leverage the strengths of convolutional layers in processing image data while incorporating the Bayesian framework to enhance diversity of the generated samples.

Therefore, the formulation of loss function for our model involves the consideration of two key loss functions outlined in Equations 1 and 11. These loss functions encapsulate the essential components for training the Bayesian DCGAN model. The discriminator loss, as expressed in Equation 12, is calculated by evaluating the difference between two main components. The first component entails a probabilistic comparison between the variational posterior and prior of the discriminator network. The second component is related to the adversarial nature of GANs and consists of the binary cross-entropy between the log probability of the real data $D(x)$ and the log probability of the generated data $D(G(z))$, Equation 12.

Likewise, the generator loss, as described in Equation 13, is determined by assessing the probabilistic distinction between the variational posterior and the prior distribution of the generator network. The second component involves the binary cross-entropy between the log probability of the generated data $D(G(z))$.

$$
\begin{aligned}
\text{Discriminator\_loss} = &\frac{1}{n}\sum_{i=1}^{n}\left(\log q(w_d^{(i)}|\theta_d) - \log p(w_d^{(i)})\right) \\
&- (\log(D(x, w_d)) + \log(1 - D(G(z, w_g), w_d)))
\end{aligned}
\tag{12}
$$

$$\text{Generator\_loss} = \frac{1}{n} \sum_{i=1}^{n} \left( \log q(w_g^{(i)}|\theta_g) - \log p(w_g^{(i)}) \right)$$
$$- \log(D(G(z, w_g), w_d)) \tag{13}$$

The "D" and "G" in equations 12 and 13 represent the discriminator and the generator network, respectively. Consequently, the pseudo-code for the learning process is structured around iteratively optimizing the parameters of both the discriminator and generator networks based on their respective loss functions, Algorithm 1.

---

**Algorithm 1** Learning procedure for Bayesian DCGAN

---

**Require:** : m, batch size
1: **for** number of training iterations **do**
2:     **for** $k$ steps **do**
3:         Sample minibatch of $m$ noise samples $\{z^{(1)}, ..., z^{(m)}\}$ from noise prior distribution $p_a(z)$.
4:         Sample minibatch of $m$ examples $\{x^{(1)}, ..., x^{(m)}\}$ from data distribution $P_{\text{data}}(X)$.
5:         Sample $\epsilon \sim \mathcal{N}(0, I)$
6:         Calculate $w = \mu + \log(1 + \exp(\rho)) \odot \epsilon$
7:         Discriminator training:
8:             Calculate loss as 12 for the discriminator network
9:             Calculate the gradient of discriminator's loss with respect to $\mu_d$ and $\rho_d$:
10:                $\Delta\mu_d = \frac{\partial \text{loss}(w_d, \theta_d)}{\partial w_d} + \frac{\partial \text{loss}(w_d, \theta_d)}{\partial \mu_d}$
11:                $\Delta\rho_d = \frac{\partial \text{loss}(w_d, \theta_d)}{\partial w_d} \frac{\varepsilon}{1 + \exp(-\rho_d)} + \frac{\partial \text{loss}(w_d, \theta_d)}{\partial \rho_d}$
12:             Update the discriminator
13:         Generator training:
14:             Calculate loss as 13 for the generator network
15:             Calculate the gradient of generator's loss with respect to $\mu_g$ and $\rho_g$:
16:                $\Delta\mu_g = \frac{\partial \text{loss}(w_g, \theta_g)}{\partial w_g} + \frac{\partial \text{loss}(w_g, \theta_g)}{\partial \mu_g}$
17:                $\Delta\rho_g = \frac{\partial \text{loss}(w_g, \theta_g)}{\partial w_g} \frac{\varepsilon}{1 + \exp(-\rho_g)} + \frac{\partial \text{loss}(w_g, \theta_g)}{\partial \rho_g}$
18:             Update the generator
19:     **end for**
20: **end for**

---

## 4 Experiments

In this section, we evaluate the performance of our proposed Bayesian DCGAN model using the MNIST dataset (Gangaputra; Qiao , 2007), a standard benchmark in machine learning. Our experiments are designed to answer the following research questions:

**RQ1:** How does the Bayesian DCGAN compare with the traditional DCGAN in terms of loss and convergence behavior?

**RQ2:** How do the generated images differ in terms of diversity?

**RQ3:** How does the inclusion of images generated by Bayesian DCGAN impact the performance of a simple neural network model compared to traditional training methods?

### 4.1 Dataset and Preprocessing

The MNIST dataset, a benchmark in the field of machine learning, comprises handwritten digits ranging from 0 to 9. Widely employed for diverse learning tasks, it is specifically crafted for image classification,

serving as a foundational dataset for the development and enhancement of prediction models, especially convolutional deep neural networks. This dataset includes $60,000$ training images and $10,000$ test images, with each digit being centrally positioned in a grayscale image of dimensions $28 \times 28$. For our study, the images were resized to a grayscale format with dimensions of $64 \times 64$ pixels, and Gaussian noise was added to the entire training dataset.

## 4.2 Model Architecture

The architecture of the Bayesian DCGAN used in the present study is illustrated in Figure 4. This architecture incorporates Bayesian Convolutional 2-dimensional and Bayesian Convolutional 2-dimensional transpose layers within a Bayesian framework to enhance robustness and diversity.

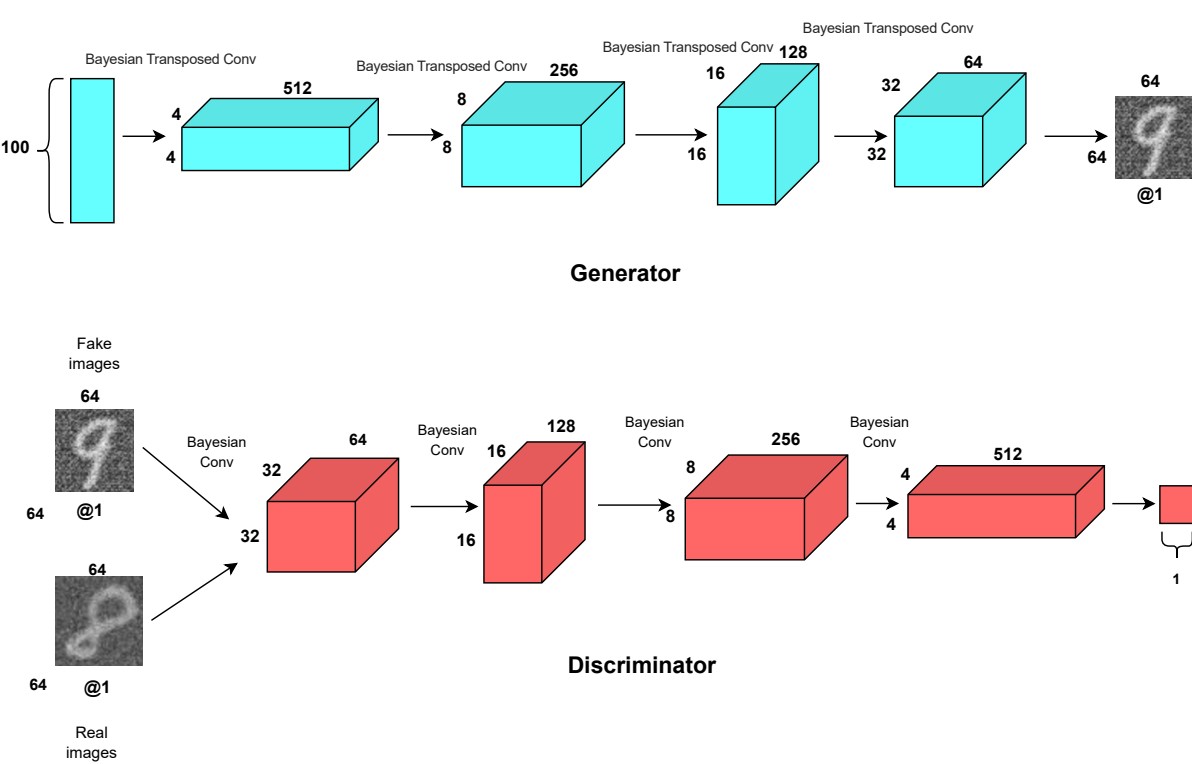

Figure 4: Bayesian DCGAN architecture used in the present study

## 4.3 Comparative Analysis of Loss Dynamics

To assess the performance of our proposed Bayesian DCGAN model, we conducted a comprehensive comparison with the standard DCGAN architecture. Figure 5a & 5b depict the generator and discriminator loss across different iterations, providing insights into the training process, convergence patterns, and the alignment of our Bayesian DCGAN model with the traditional DCGAN.

As it was mentioned, the generator loss measures how well the generator is performing in generating realistic samples, while the discriminator loss assesses the ability of the discriminator to differentiate between real and generated samples. In the initial stages, both losses fluctuate as the model undergoes learning. Subsequently, the generator loss demonstrates a downward trend, reflecting an improvement in the generator's ability to produce realistic samples. The Bayesian DCGAN exhibits reduced fluctuations in the generator loss throughout the training process compared to the traditional DCGAN. The convergence of both losses indicates that the Bayesian DCGAN model has reached a certain optimum, suggesting that further improve-

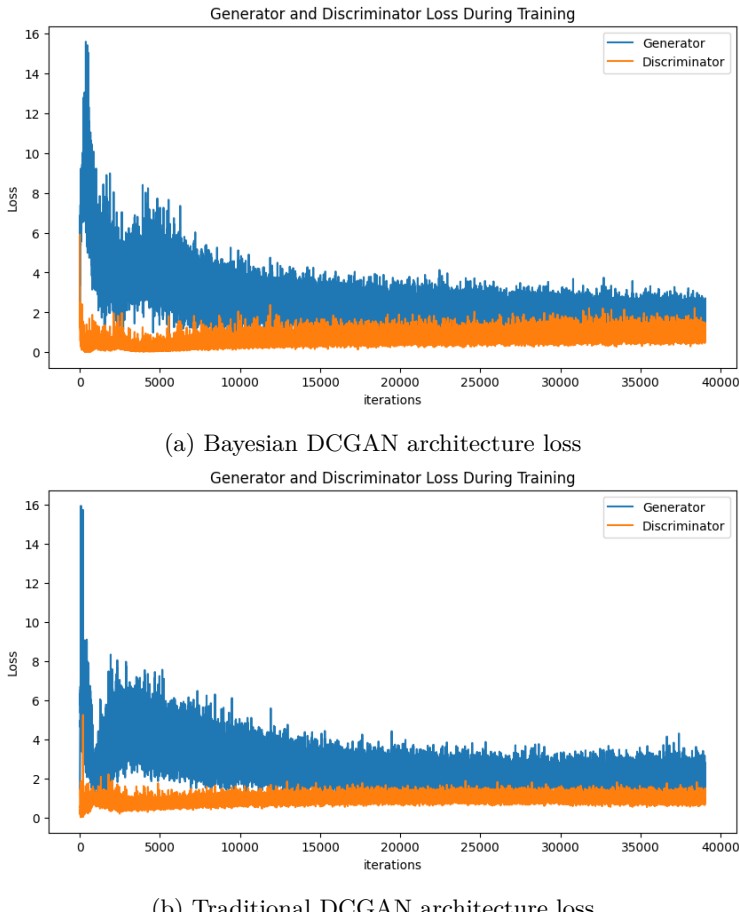

(a) Bayesian DCGAN architecture loss

(b) Traditional DCGAN architecture loss

Figure 5: The generator and discriminator loss during training for Bayesian DCGAN and Traditional DC-GAN.

ment is limited. This convergence also signifies that the model has acquired a sufficient level of learning and proficiency in generating and discriminating samples.

Figure 6a & 6b showcase a selection of randomly generated MNIST digit samples produced by both the traditional DCGAN and the Bayesian DCGAN models. Notably, the Bayesian DCGAN model exhibits the ability to generate images that closely resemble those from the real dataset. This demonstrates the model's enhanced capacity to capture and replicate the intricate patterns and features present in the original MNIST dataset, emphasizing the efficacy of the Bayesian approach in improving generative performance.

## 4.4  Diversity Analysis of Generated Samples

To verify the diversity of the samples, we employed a rigorous mathematical analysis aimed at quantifying the diversity inherent in the images generated by Bayesian DCGAN and contrasting them with those generated by traditional DCGAN. The methodology involves the creation of two distinct datasets, each consisting of $32,000$ images produced by their respective GAN algorithms. Each image within these datasets possesses dimensions of $64 \times 64$, amounting to a total of 4096 pixels, and is treated as a vector for the purpose of this analysis. In viewing each generated image as a vector with a dimensionality of 4096, we computed the sample covariance for each dataset, resulting in two covariance matrices of size $4096 \times 4096$. This mathematical approach enables a comprehensive understanding of the relationships and variations present among the pixels of the generated images, facilitating a quantitative assessment of diversity.

Fake Images 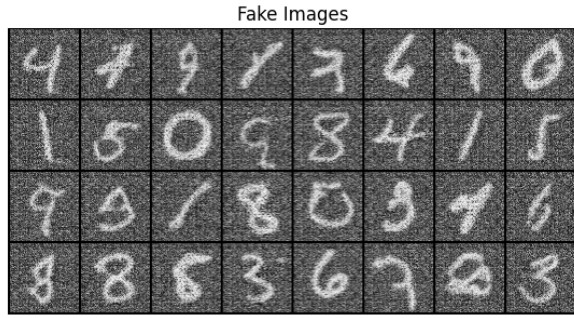   Fake Images 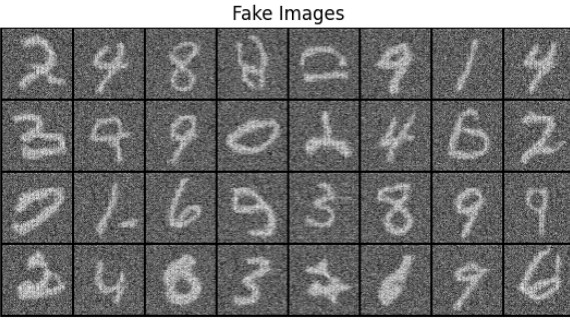

(a) Bayesian DCGAN                    (b) Traditional DCGAN

Figure 6: MNIST digits generated from random noise by Bayesian DCGAN and Traditional DCGAN.

Denoting the $32,000$ images generated by traditional DCGAN as $d_1, d_2, \ldots, d_N$ and those generated by Bayesian DCGAN as $b_1, b_2, \ldots, b_N$ (where $N = 32000$), individual components of these vectors are represented as follows:

$$d_i = (d_i^1, \cdots, d_i^{4096})$$
$$b_i = (b_i^1, \cdots, b_i^{4096})$$

The population (or sample) mean is defined as:

$$\bar{d} = (\bar{d}^1, \cdots, \bar{d}^{4096})$$

where for any $1 \leq j \leq 4096$:

$$\bar{d}^j = \frac{1}{N} \left( \sum_{i=1}^{N} d_i^j \right)$$

This definition holds for the images generated by Bayesian DCGAN as well. Further, for any $1 \leq r, s \leq 4096$, the population covariance between $r^{th}$ and $s^{th}$ components is defined as:

$$Cov(d^r, d^s) = \frac{1}{N} \left( \sum_{i=1}^{N} (d_i^r - \bar{d}^r)(d_i^s - \bar{d}^s) \right)$$

This computation results in the $(r, s)$ entry of the population covariance matrix for the data generated by Bayesian DCGAN, and a similar definition can be applied for traditional DCGAN. The criteria for diversity is the eigenvalues of the covariance matrix. Suppose

$$\lambda_1 \geq \lambda_2 \geq \cdots \geq \lambda_{4096}$$

are the sorted eigenvalues of the traditional DCGAN population covariance, and

$$\mu_1 \geq \mu_2 \geq \cdots \geq \mu_{4096}$$

are those of Bayesian DCGAN. From our experiment, it was observed that:

$$\lambda_1 < \mu_1$$

$$\lambda_2 < \mu_2$$

.

.

$$\lambda_{4096} < \mu_{4096}$$

Table 1 shows first six terms of the $\mu$ and $\lambda$. The observation that the claim holds for the first six terms, and, in fact, for all indices $j$ (i.e., $\mu_j > \lambda_j$), provides compelling evidence supporting the assertion that Bayesian DCGAN exhibits greater diversity when compared to traditional DCGAN. This comparison is based on the eigenvalues of their corresponding population covariance matrices. The consistently larger eigenvalues of the Bayesian DCGAN matrix signify a greater covariance matrix, indicating a broader spread and increased variability in the generated samples. Therefore, the mathematical analysis strongly suggests that the Bayesian DCGAN model produces a more diverse set of samples than traditional DCGAN.

Table 1: Comparing Eigenvalues Across Different Models

| Bayesian DCGAN | Traditional DCGAN |
|---|---|
| $\mu$ | $\lambda$ |
| 26.298 | 11.399 |
| 18.938 | 9.089 |
| 16.208 | 7.628 |
| 14.185 | 6.457 |
| 11.764 | 5.500 |
| 11.603 | 4.990 |

## 4.5   Impact on Model Performance

To validate our findings, we compared two models: one trained solely on a 10% subset of the MNIST dataset and another trained on a combination of a 10% subset of the MNIST dataset and approximately 1400 images generated by Bayesian DCGAN. Both models were evaluated on unseen images from Bayesian DCGAN, and the results are presented in Table 2. Interestingly, even when trained on a small amount of generated data, the model exhibited better performance and greater generalization. On the other hand, in the pursuit of reducing uncertainty in neural networks, deep ensembles have gained significant attention. This method involves training neural networks from scratch multiple times on the dataset, resulting in distinct models. During testing, the dataset is passed through each model, and the final output is obtained by averaging their outputs. Ensemble strategies vary in their approaches to selecting baseline classifiers for training, with two primary types: homogeneous and heterogeneous ensembles. While homogeneous ensembles utilize the same type of baseline classifiers trained on different data subsets, heterogeneous ensembles employ various types of classifiers trained on the same dataset (Mohammed & Kora, 2023). In our study, we chose heterogeneous ensembles as the baseline method for uncertainty reduction (Table 2). Notably, the model trained on a combination of MNIST and generated images outperformed ensemble learning models. It is worth mentioning that we employed a simple model with only two convolutional layers and trained it for a few epochs to facilitate result comparison.

Table 2: Comparison of different models

| Trained on MNIST | Ensemble learning on MNIST | Trained on MNIST and generated images |
|---|---|---|
| 82% | 83.6% | 86% |

## 5   Conclusion

This study focused on addressing mode collapse in DCGAN and enhancing sample diversity. This approach involves the introduction of a Bayesian framework applied to DCGAN, referred to as Bayesian DCGAN. Within this framework, a weight distribution is incorporated into the network using the Bayes by Backprop method. Additionally, a mathematical approach is presented to evaluate the diversity in samples generated

by Bayesian DCGAN, comparing it with the outcomes of conventional DCGAN. The experimental findings reveal that Bayesian DCGAN consistently produces larger eigenvalues, indicating a broader spread and variability in the generated samples compared to its conventional counterpart. This signifies a significant reduction in mode collapse within DCGAN and highlights the effectiveness of the Bayesian DCGAN framework in generating more diverse samples. While this study effectively tackled mode collapse in DCGAN and demonstrated the efficacy of Bayesian DCGAN in improving sample diversity, there are several possibilities for further exploration and improvement. This includes investigating ways to integrate domain-specific constraints or additional information into the Bayesian DCGAN framework. These constraints could pertain to specific data distributions, allowing the model to generate samples that conform to particular desired characteristics.

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
