# OpenReview forum: "Enhancing Sample Diversity in Deep Convolutional GANs Using a Bayesian Framework"
_TMLR — Rejected by TMLR_

### Review · Reviewer_sn9W · 2024-10-09

**Summary Of Contributions:**

This paper employed mean-filed variational inference as the posterior approximation technique for Bayesian convolutional generative adversarial neural network (Bayesian DCGAN) to improve the diversity of generated samples, addressing the mode collapse issue in DCGAN. A numerical comparison between the proposed methods and conventional point-estimate (non-Bayesian) training of DCGAN is conducted on the MNIST dataset, demonstrating an improved generated sample diversity based on the eigenvalues of the sample covariance matrix.

**Audience:**

Yes

**Broader Impact Concerns:**

No ethical implications of the work are of concern.

**Claims And Evidence:**

No

**Requested Changes:**

1. Explicitly demonstrate the algorithmic difference between the proposed method and the Bayesian DCGAN in Saatci & Wilson (2017).
2. Compare the proposed method with the Bayesian DCGAN in Saatci & Wilson (2017) and explain why the modification(s) in 1., e.g., HMC to mean field variational inference is necessary or beneficial.
3. In this context, state more accurately the contributions of this paper compared to existing methods for improving sample diversity and Bayesian GAN.
4. Include other methods that either address the mode collapse problem or adopt the Bayesian framework to GAN, and conduct experiments on more challenging datasets, in terms of both sample diversity and model performance.
5. Explain why the proposed method is limited to DCGAN, or discuss whether it can be extended to general GAN.

**Strengths And Weaknesses:**

**Strengths**

1. The paper is overall well-written.
2. The evaluation metric of sample diversity based on eigenvalues is somewhat new.

**Weaknesses**
1. The idea of leveraging Bayesian deep learning to improve sample diversity of generative models is extensively explored since Saatci & Wilson (2017). In their section 4.2, the DCGAN variant of their Bayesian GAN is even called Bayesian DCGAN, identical to the method proposed in this paper. Thus the reviewer finds the first contribution of this paper over-claiming.
2. The only difference the reviewer finds between this paper and Saatci & Wilson (2017) is the way to estimate the posterior distribution, where Saatci & Wilson (2017) applied stochastic gradient Hamiltonian Monte Carlo (HMC) while this paper applies mean field variance inference. However, the motivation for this replacement has not been introduced, e.g., why this change is necessary/beneficial.
3. The numerical experiment is simple. While there are extensive existing methods that also address the mode collapse problem, e.g., Unrolled Generative Adversarial Networks (Metz et. al., 2017) and its many citations, the paper only compares with the vanilla DCGAN, not to mention the lack of comparison with the Bayesian DCGAN in (Saatci & Wilson 2017) and the simplicity and limited variety of the dataset.

----

References:

Saatci, Yunus, and Andrew G. Wilson. "Bayesian gan." Advances in neural information processing systems 30. 2017.

Metz, Luke, et al. "Unrolled Generative Adversarial Networks." International Conference on Learning Representations. 2017.

---

### Review · Reviewer_ragc · 2024-10-11

**Summary Of Contributions:**

The paper presents a Bayesian DCGAN framework aimed at addressing mode collapse by introducing a probability distribution over network weights, using the Bayes by Backprop method. The main claim is that Bayesian DCGAN produces higher sample diversity than standard DCGAN.

**Audience:**

Yes

**Broader Impact Concerns:**

I have no concerns here.

**Claims And Evidence:**

No

**Requested Changes:**

- Clarify the novelty of the Bayesian approach over prior work and whether the proposed mean-field variational inference offers tangible benefits.
- Expand experiments to include additional datasets and use standard diversity metrics like FID to substantiate claims regarding sample diversity.
- Provide a quantitative comparison between Bayesian and conventional DCGANs, particularly focusing on performance and sample diversity.
- Revise the writing to remove redundancy and ensure technical accuracy.

**Strengths And Weaknesses:**

**Strengths**
- The topic of mode collapse in GANs is relevant.

**Weaknesses**
While the topic of mode collapse in GANs is relevant, several issues arise in the execution, novelty, and experimental validation of the claims.  The paper lacks rigorous experimental validation to substantiate its claims. It also claims questionable novelty. Significant revisions are required to clarify the contribution, correct the mathematical notations, and provide a more comprehensive experimental analysis. Currently the manuscript is very far from publication standards. Details are listed below.

1. **Insufficient Experimental Validation:**
    - The claim that the Bayesian DCGAN produces more sample diversity is unconvincing. The current metric of sample diversity, based on the covariance matrix of generated samples, could be skewed by noise, rather than reflecting genuine diversity.
    - Standard diversity metrics such as FID should be used to provide a rigorous and widely accepted measure of sample diversity.
    - The study relies solely on MNIST, which is too simple a dataset to robustly test the effectiveness of the method. Additional experiments on more complex datasets like CIFAR-10 or Celeb-A are necessary with standard metrics like FID.
2. **Questionable Claims of Novelty:**
    - The application of Bayesian neural networks (BNNs) to GANs is not novel. Prior work such as BayesGAN (Saatchi et al., NeurIPS 2017) has already applied Bayesian methods to GAN architectures. In principle that’s of course okay. But the paper does not clearly articulate how its approach offers advantages over previous work, especially in comparison to MCMC-based methods.
3. **Convergence and Model Performance:**
    - While the authors claim that the Bayesian DCGAN reduces generator loss fluctuations, there is no quantitative analysis. From the provided loss function plots I can barely discern the claim. A quantitative analysis is needed to support such a claim.
    - There is no comparison between Bayesian DCGAN and conventional DCGANs in  section  performance section 4.5 (Impact on model performance). A comparison of models trained on GAN produced datasets would be natural especially given the claim of superior sample diversity. The fact that augmenting the 10% of MNIST with Bayesian DCGAN samples produces 3 % accuracy increase is not surprising. This would also happen with a DCGAN and I suspect it would not be much less of an accuracy gain given that it was not reported.
4. **Redundancy and Writing:**
    - The paper contains several instances of redundancy, particularly when explaining the Bayesian framework and the concept of weight distributions.
    - There are minor issues with the writing, such as phrases like "it is worth mentioning," which detract from the overall clarity and flow of the paper. Everything on a paper is worth mentioning. The writing would benefit from refinement.
5. **Misinterpretation of Concepts:**
    - Equation 11 is wrong. The law of large numbers does not imply that the expectation equals the sample average, but rather that the sample average converges to the expectation in the limit.
    - Some mathematical notations, such as in Equation 12, are incorrect and need to be revised for clarity.

---

### Review · Reviewer_TxLp · 2024-10-12

**Summary Of Contributions:**

This paper aims to improve Bayesian Deep Convolutional Generative Adversarial Networks (DCGANs) with greater sample diversity. Specifically, the authors applied a mean-field variational inference approximation to calculate the evidence lower bound (ELBO). The authors further examine their method on the MNIST dataset.

**Audience:**

Yes

**Broader Impact Concerns:**

No.

**Claims And Evidence:**

No

**Requested Changes:**

1. Please consider demonstrating more differences between the submitted manuscript and [1].

2. The reviewer will be interested to see a better baseline study on more datasets.

3. Please improve the design of Figure 3 and use an original design for Figure 2.


[1] Blundell, Charles, et al. "Weight uncertainty in neural network." International conference on machine learning. PMLR, 2015.

**Strengths And Weaknesses:**

Overall, this paper provides a good overview of the Bayesian GANs and improves the problem of mode collapse, which is often seen in GANs. However, the quality of the paper can be further improved (e.g. the figures, experiments), and the reviewer is interested in seeing a better presentation of the improvements (from the traditional GANs).

1. Increasing the diversity of generated images is an interesting research question as the training of GANs is known to be complex.

2. Using the mean-field variational approximation is a practical method to resolve this problem with advantages computationally.

The weaknesses of this paper are also clear.

1. The paper has very similar plots (with lower quality), derivations and experiments compared to [1]. It is hard for the reviewer to understand whether the contribution is from [1] or from the submitted manuscript.

2. The experimental study of this paper is very weak, lacking an extensive baseline study. The reviewer would wish to see more evidence of the advantages of the proposed method.

3. The training of GANs is complex, making it difficult to pinpoint where the improvement comes from. This affects the reliability of the presented results.

References:

[1] Blundell, Charles, et al. "Weight uncertainty in neural network." International conference on machine learning. PMLR, 2015.

---

### Decision · Action_Editor_ts9C · 2024-12-29

**Recommendation:** Reject

**Comment:**

Reviewers point out a few major issues, such as the lack of clarifications on the difference with a prior work, and insufficient experiments to support the main claims. Reviewers all recommend rejection. The authors did not provide responses to the comments. So I recommend rejection.

**Audience:**

The paper is quite similar to Baysian DCGAN. The reviewers asked for clarification on the similarity, and there is no response from the authors.   The similarity with prior work makes it unclear whether some audience would be interested in this paper.

**Claims And Evidence:**

This work does not suport the claims made in the paper, as pointed out by reviewers. The paper claims that its approach produces more sample diversity, but the current metric of sample diversity based on the covariance matrix could be skewed by noise.  A reviewer suggests to add experiments with standard metric like FID to support claims on sample diversity, and no such experiment is provided.  In addition, the claim that Bayesian DCGAN reduces generator loss fluctuations is not observed by the presented plots, and a reviewer requests additional experiments or quantitative analysis. Again, there is no response from reviewers on this request. Overall, the claims of the paper are not supported by enough evidence.